# Characteristics and Challenges for the Development of Nature-Based Adult Day Services in Urban Areas for People with Dementia and Their Family Caregivers

**DOI:** 10.3390/ijerph16081337

**Published:** 2019-04-14

**Authors:** Jan Hassink, Lenneke Vaandrager, Yvette Buist, Simone de Bruin

**Affiliations:** 1Wageningen Plant Research, Agrosystems Research, Wageningen University& Research, P.O. Box 16, 6700AA Wageningen, The Netherlands; 2Department of Social Sciences, Health and Society, Wageningen University & Research, P.O. Box 8130, 6700EW Wageningen, The Netherlands; lenneke.vaandrager@wur.nl; 3Center for Nutrition, Prevention and Health Services, National Institute for Public Health and the Environment, P.O. Box 1, 3720 BA Bilthoven, The Netherlands; yvette.buist@wur.nl (Y.B.); simone.de.bruin@rivm.nl (S.d.B.)

**Keywords:** nature-based services, people with dementia, social entrepreneurs, cities

## Abstract

Nature-based adult day services (ADSs) for people with dementia (PwD) are well-known in rural areas. In recent years, a number of providers have started offering these services in urban contexts, e.g., in city farms and community gardens, where people with dementia participate in outdoor activities, such as gardening and caring for animals. At the moment, little is known about these services within an urban context, and the aim of this study is to characterize different types of nature-based ADSs in urban areas for PwD living at home, as well as to identify general and specific challenges with regard to the development of different types of ADSs. An inventory was carried out and 17 ADS providers in urban areas were interviewed about their initiatives, settings, client groups, motivations for starting their nature-based ADS, and their experiences with, competences for, and funding of urban nature-based ADS. The interviews were recorded, transcribed verbatim and thematically analyzed. Five types of nature-based ADSs were identified: (1) services offered by social entrepreneurs, (2) nursing homes opening their garden to people with dementia, (3) social care organization setting up nature-based, (4) community garden set up by citizens, and (5) hybrid initiatives. Common activities were gardening, preparing meals, and taking care of farm animals. The main activities organized by nursing homes included sitting and walking in the garden and attending presentations and excursions. General challenges included the availability of green urban spaces and acquiring funding for the nature-based services. Initiatives of social entrepreneurs depended strongly on their commitment. Challenges for nursing homes included a lack of commitment among nursing staff, involvement of PwD living at home and a lack of interaction with the neighborhood. Volunteers played a key role in the initiatives organized by social care organizations and in community gardens. However, it was a major challenge to find volunteers who know enough about care and gardening. Specific challenges for the hybrid types were related to differences in work culture between social entrepreneurs and care organizations. Different types of care-oriented and community-oriented nature-based adult day services in urban areas for people with dementia have been developed, facing different types of challenges. Care oriented initiatives like nursing homes opening their garden focus on risk prevention and their nature-based services tended to be less diverse and stimulating for people with dementia living at home. Collaboration between such care-oriented initiatives with initiatives of other types of organizations or social entrepreneurs can lead to more appealing community-oriented nature-based services.

## 1. Introduction

Dementia is an important public health issue. It is estimated that, there are currently between 254,000 and 270,000 people with dementia in the Netherlands, which is expected to double by 2040 [1]. People suffering from dementia are encouraged to live at home for as long as possible, to allow them to participate in society with support from their network and local services, as an alternative for the more expensive and institutional care [2,3]. An important challenge related to this is the burden it places on family caregivers. Although adult day services (ADS) are important facilities to relieve part of that burden [2], less than 10% of people with dementia take part in these services [1]. ADS centers often do not match the needs of people with dementia, making them reluctant to take part in these services and generating feelings of guilt among family caregivers [1]. There is an increasing interest in the use of outdoor areas and activities for people with dementia, as they offer opportunities for active engagement with plants and animals [4]. Green care farms that offer nature-based ADS providing these opportunities distinguish themselves from conventional adult day service centers by their non-institutional, homelike character, diversity in activities, opportunities for social interaction and spending time outdoors, and by providing a healthy diet [5,6]. They can also provide relief and reduce feelings of guilt among family caregivers [5,6,7]. Because most people with dementia live in urban areas and transporting them to green care farms, which are usually situated in rural areas is often challenging and costly, it seems worthwhile to establish nature-based services for people with dementia in urban areas as well [3]. 

In recent years, several nature-based services have been established in an urban context, e.g., at city farms and community gardens, often initiated by social entrepreneurs and including the involvement of citizens [8,9]. Information about these initiatives and the use of urban green space in care provision is still scarce [10]. Learning more about the organizational characteristics and key factors affecting the development of these nature-based day services for people with dementia in urban areas is useful for a number of reasons: (i) nature-based ADSs in urban areas may provide an attractive alternative to rural green care farms, because they provide easier access to people with dementia, and they share similar values and benefits as green care farms [11]; (ii) They have been advocated as part of the healthy ageing approach by the European Healthy Cities Network [12] which promotes personal and communal empowerment, access to services, and supportive physical and social environments [12].

The aims of this study are twofold:Provide insight into the characteristics of different types of nature-based ADSs in urban areas for people with dementia living at home. Explore common and specific challenges for the development and running of different types of nature-based ADSs in urban areas.

The characteristics of nature-based ADS can be affected by a number of factors. In a recent study, Hassink et al. [13] described how the initiator’s background affects the organizational characteristics of a green care farm and their strategy to develop the care farm. Social and entrepreneurial literature also pointed to the importance of the background of entrepreneurs in new organizations [14,15], in relation to the locus of entrepreneurship: who is the initiator of the nature-based initiative and where does the inspiration originate. Access to financial resources, continuous and varied public relations activities, cooperation with other local organizations and the added value of the initiative to the organization appear to be key factors in the development of innovative services involving care for people with dementia [16]. Furthermore, Lourida et al. [17] identified time constraints and workload as the most important organizational challenges. Leadership and managerial support were described as facilitating factors, mainly with regard to the development of practices in residential dementia care [17]. 

When different types of initiators together initiate a nature-based ADS, working together can be a challenge. Collaboration between social entrepreneurs and institutional care providers was found to be challenging due to differences in dominant logics, institutional barriers, a lack of legitimacy, and a lack of commitment [18]. In some cases, the parties involved had different ideas about quality and safety issues. Committed and strategically operating boundary spanners and support from top-level management were found to be crucial to a successful collaboration and implementation of green care services [18]. 

## 2. Methods

### 2.1. Study Design

This study was a mixed methods study in which both quantitative and qualitative data were collected on nature-based ADS. The study consisted of two phases. In the first phase, a national inventory of nature-based ADSs in urban areas in the Netherlands was conducted. In the second phase, initiators of the ADSs identified in the first phase were invited to participate in an interview, with the aim of gaining a deeper understanding of the characteristics of their initiatives and of the challenges that they faced when setting up their service. This study is a qualitative inquiry, using semi-structured interviews with the initiators of different nature-based ADSs in urban areas. This qualitative inquiry is particularly useful for exploratory studies, when the object is to gain a holistic understanding of how dynamics unfold in real life settings [19].

### 2.2. Inventory of Nature-Based ADSs in Urban Areas for People with Dementia Living at Home

The initiators of nature-based ADSs were recruited by placing an invitation to participate in the study on several Dutch websites (including the website of a Dutch patient organization for people with dementia and a website about nature-based services) and in several newsletters on dementia care and nature-based services. Additionally, they were actively approached through the researchers’ networks. They were asked to complete an online survey. In total 21 representatives of nature-based ADSs completed the survey. 

### 2.3. Interviews with Initiators of Nature-Bases ADSs

In total 15 of the identified initiators were approached for an interview. Some initiators (n = 6) were not invited, because their initiatives appeared to be still in the start-up phase and/or did not yet provide services to people with dementia. In two cases, the initiators were interviewed in pairs: in one case, the manager of the care organization and a social entrepreneur, and in another case, a social entrepreneur and the person responsible for the day care activities for people with dementia. In total, 17 people were interviewed. 

The respondents were visited at the location of the nature-based service for face-to-face interviews that lasted approximately an hour and were carried out by the first and fourth author and an assistant researcher. The interviews were audio taped and transcribed verbatim. 

In line with the objectives the interviews covered the following topics: 1. Background and motivation of initiators for setting up the nature-based ADS and description of the activities; 2. Characteristics (location, activities, type and number of participants, quality and safety issues) of the nature-based service; and 3. Challenges for the development and running of the initiative and key factors affecting its implementation. 

### 2.4. Data Analysis

The data analysis used was an inductive, iterative process, as proposed by Strauss and Corbin [20]. First, all transcripts were read. Instead of using a predetermined coding scheme, themes that were related to the topics of the interviews could emerge from the participants’ own words, as recommended in exploratory research [20]. A constant comparative method was used to code and analyze the data simultaneously and categorize the information into developing themes representing recurring patterns of behavior and meaning. Once the themes had been identified, the data were mined for elements related to those themes. The data were analyzed by the first two authors. To organize the coded transcripts and sort the data according to the themes, a computer program for qualitative data analysis (MAXQDA 2018 (Verbi GmbH, Berlin, Germany) was used. After the interviews were coded and analyzed, the findings were discussed by the four authors.

## 3. Results

### 3.1. Characteristics of the Different Types of Nature-Based ADSs

Based on the inventory and the interviews, five main types of initiatives were identified.Social entrepreneurs offering nature-based ADSs, either byUsing their own facilitiesParticipating in existing facilities (e.g., city farm, city garden, park);Nursing homes opening their gardens to people with dementia living at home; Social care organization setting up nature-based activities (e.g., green maintenance, walks in green environments, visit to a children’s farm or city farm); Community garden set up by citizens;Hybrid initiatives: Care organizations initiating nature-based ADSs together with other actors:Either other institutional partnersOr social entrepreneurs.

In the following paragraphs the main characteristics of the different types of initiatives are described, focusing on the background and motivation of the initiators and the characteristics of the facilities: the type of location, activities, participants and issues concerning quality and safety. Table 1 provides an overview of the characteristics of the different types of initiatives.

#### 3.1.1. Social Entrepreneurs Offering Nature-Based ADS

The backgrounds of the initiators of this category were diverse. Of the seven respondents who classified themselves as social entrepreneurs, three had a background in healthcare as well as agriculture, two had a nature-based background (landscape architecture and nature education) and saw an opportunity to use urban green areas to provide healthcare-related services, one was a health care professional who moved to a green area and decided to start her own business, giving her the freedom to decide how to provide care and shape the ADS. Finally, a former international development worker who became a passionate gardener and wanted to change his career. 

Social entrepreneurs initiated the nature-based ADSs since they experienced a desire to provide an alternative to institutional day care services and they realized that there are insufficient opportunities for an increasing number of elderly to take part in meaningful activities. The social entrepreneurs shared the passion to combine care and nature.
“As an advisor and caregiver, I have spent years on initiatives for people with dementia in relation to a green environment. Only this time, I do it myself. I made it my own enterprise. It is a kind of personal passion” 

Differences were found between the initiatives in terms of locations and nature-based activities (Table 1). Some social entrepreneurs initiated their own city farm or nursery, while others decided to work together with existing green facilities, such as a city farm or a community garden. The initiators were motivated to activate people with dementia and provide them with useful tasks. Typical activities for people with dementia include growing and harvesting vegetables and flowers, taking care of the animals, preparing a meal together, creative activities, making raised beds, crafting wood, and walking. 

In addition, there were differences in the number and diversity of participants in the nature-based ADS (Table 1). The number of people with dementia living at home taking part in the nature-based day services initiated by social entrepreneurs ranged between 3 and more than 30 per day. Most initiatives were open five days a week for people with dementia, while the two initiatives using existing locations were open to people with dementia one or two days a week. On the other days, the locations were reserved for other activities. Two of the social entrepreneurs developing their own nature-based ADS specialized in dementia care and only developed day activities for people with dementia living at home. The other social entrepreneurs deliberately decided to provide their services to a broader range of participants such as adults with mental problems and intellectual disabilities. They wanted to create a diverse setting that is a mirror of society, reduce the stigma attached to dementia and create a more informal atmosphere, and allow people with different backgrounds to stimulate and help each other.
“The location is a mirror of society, that it has the same diversity. It will provide you with positive stimuli, instead of thinking ‘oh, I am part of that group’. The beautiful surroundings, the green environment, can address individual needs. Not with the group, but simply in smaller groups.” 
“For instance, we had a participant with acquired brain injury, who is the supervisor in such a group of people with dementia. Which means that both benefit from the activities.” 
“And when a new person comes and takes a look, they can’t see who is a patient or caregiver. So that distinction becomes less clear and, for some, that is very beneficial.” 

On all locations, a considerable number of volunteers and students (interns) were involved.
The more varied the group is, the better it actually works. I was very reluctant with interns, but I am completely convinced now. Because we have had some healthcare interns for a kind of exploratory internship. They were young and enthusiastic, and they all had that social talent, except for one. Well, they brought a lot. They bring their stories, their entire lives, you know? And a boy from a green course, who was slightly autistic, he was an intern here. Well, he liked it so much that he comes back every vacation. And the people who remember look forward to his arrival. So the more varied, the better. 

#### 3.1.2. Nursing Homes Opening Their Gardens to People with Dementia Living at Home

The three initiators of nature-based ADSs provided by nursing homes were all managers or general coordinators with responsibility for ADSs. The nursing homes opened up their gardens to people with dementia living at home. The initiators involved were unhappy about the way the gardens had been used up to that point: many gardens were not really accessible to or used by people with dementia. In two cases, initiating nature-based services matched with the vision of the care organization. One of the managers indicated that their care organization provides care for many clients with a rural background. Therefore, she indicated that nature and being outside should be important elements of day activities.

In these three nursing homes, the people with dementia were not expected to take part in the maintenance of the garden. They mainly sat in the garden and were able to enjoy the green environment. Additional activities for people with dementia were also organized, such as excursions, presentations, the visit of a beekeeper or a joint walk.

When nature-based ADSs were established in nursing home gardens, most of the participants were people with dementia living in the institution, while the number of people with dementia living at home and other people from the neighborhood or volunteers tended to be limited. 

#### 3.1.3. Social Care Organization Setting Up Nature-Based Activities

The social care organization had several ADS centers where general activities for people with dementia living at home are organized, such as drinking coffee and playing games. Some of the participants were also interested in taking part in other activities. The manager initiated a collaboration with a city farm near the day center. The city farm was interested in working together, which was in line with its mission to bring people into contact with farming activities and farm animals. This allowed the manager to develop a program where participants start at the day center. After coffee, a volunteer joins participants interested in visiting the city farm, while the other participants stay at the ADS center. 

The initiator of the social care organization was a former dementia case manager who recognized the needs of people with dementia to participate in productive activities and the desire of some of the persons with dementia to work outside.

The activities of people with dementia participating in the nature-based ADS of the social care organization on the city farm were limited. Under the guidance of a volunteer, they visited the city farm for 1–2 h. The activities involved feeding some of the chickens and rabbits, sitting on the terrace watching the other visitors. 

The number of people with dementia participating in the nature-based services offered by the social care organization was limited (n = 3) and the services were only available to people with dementia for one part of the day per week. People with dementia were supported by volunteers and the interaction between the participants and visitors of the city farm was limited. 

#### 3.1.4. Community Garden Set Up by Citizens 

This initiative was started by volunteers with the ambition to upgrade an untidy green location in their neighborhood which generated aggression and vandalism. The key initiator was a retired psychotherapist and a former director of a psychiatric hospital with good contacts with the municipality. He started a foundation.

The major motivation of the initiative was to increase the quality of a green location in the neighborhood and to make it a pleasant meeting place for people in the neighborhood. The initiators were very concerned with stimulating the participation of vulnerable citizens in their neighborhood.
“And then I saw this mess and I thought that is not a good place for people to be in. I then set up the plan to start a care garden here” 

The volunteers created a community garden, the underlying philosophy being that people with dementia should enjoy the garden and not feel obliged to engage in any activity. The initiator indicated that the people with dementia mainly visit the garden to enjoy the green environment and meet other people. 

The number of people with dementia taking part in the community garden was limited (n = 3) and the garden was only open for a few days per week. Volunteers facilitated the day activities for the people with dementia. 

#### 3.1.5. Hybrid Initiatives 

The hybrid types of ADSs consisted of initiatives where care organizations initiated the nature-based service together with other parties, such as a building company, a childcare organization, an organization for people with learning disabilities and an voluntary organization (Type 5a Table 1). The managers who developed nature-based ADSs together with other organizations stressed the importance of connecting and making their location accessible to the people in the neighborhood.
“The initiative was also designed to connect people. Get them to relax, away from the medical culture, so to speak” 

In one case of Type 5a, the initiator had seen how beneficial care farming had been to a client in her organization who had received day care at a green care farm. However, it was too complicated to organize transportation to a green care farm, so she looked for care farming alternatives in the city. Working together with a social entrepreneur made it possible to realize these ambitions.

The hybrid nature-based ADSs included a community garden, a greenhouse and a city farm, providing a diversity of nature-based activities with the support of green professionals and a considerable number of volunteers. 

The hybrid nature-based ADSs hosted a diversity of participants from the organizations taking part, as well as a considerable number of volunteers and people from the neighborhood, e.g., by organizing neighborhood days.

The group of people taking part in the nature-based day service initiated by the care organization and the social entrepreneur changed over time (Type 5b Table 1). Initially, when the social entrepreneur was responsible, people with dementia living at home interacted with visitors and volunteers involved in the urban agriculture project. The social entrepreneur indicated that when he left and his tasks were taken over by employees of the care organization, the interaction between people with dementia and people involved in the urban agriculture project became limited, much to his regret.
“There is less and less interaction with our volunteers and visitors. In the past, there was more laughter, and I miss that now”

### 3.2. Quality and Safety Issues

Initiators of all five types underlined the importance of the quality of the care they provide. Initiators had different ideas about good quality of care and about the importance and use of protocols to guarantee quality of care and safety. Two different approaches can be distinguished. A medical approach, focusing on safety and a community approach, focusing on participation in the community. The initiators from nursing homes and care organizations all agreed on the importance of a focus on people’s wellbeing rather than only medical care. At the same time, they all stressed the importance of quality control and made it clear that providing care is their core business. In their view, quality of care involved minimizing risks and employing professional personnel. Concrete issues included attention to poisonous plants, water in which people with dementia can fall, and the risk of stumbling. In their view, employees should have a formal education as a carer or social worker. They indicated that the way to increase quality of care is a systematic monitoring of the satisfaction of the care and multidisciplinary consultation.
“Caring is our core business of course in which we want to improve ourselves day by day”
“The staff has to include safety in all activities. For instance, regulations say we do not put poisonous plants in the corridors or in the garden. And there will be more rules like that. Falling, we must not have things that people can easily trip over. We must not have water in the garden that people can easily fall into. Those are certainly things we keep in mind.” 

Some of the initiators mentioned the tension they experience between safety and the clients’ freedom.
“We have a garden, people can move freely here. It isn’t a living room, where people sit still all day and you can keep an eye on them at all times. The added value of the location is that people have some freedom, they can enter the garden or the greenhouse. As a result, the staff cannot watch everybody all the time and people can walk into the hallway and go outside, for instance while we are talking here. That is the charm of the location and something that you need to discuss with relatives, is this something we accept?” 

The social entrepreneurs, social care and the neighborhood initiative had little affinity with formal quality control issues and protocols. Instead of this, they emphasized the importance of freedom for clients and providing a setting that is as normal as possible.
“Things have been going well for 2.5 years, without all kinds of protocols and thick reports. If there is an inspection, which I hope, for them to show up one day to have a look … For me, it’s also a learning process. Is that acceptable, is that possible? Legally and in accordance with the rules the way we are doing things. So I would like to have an inspection and show them well, we are working without protocols, what do you think about it?”

Also, the social entrepreneurs and the initiator of the neighborhood initiative underlined the importance of providing quality care and meeting the demands of people with dementia and their primary caregivers. For them, direct and personal contact with the participants and primary caregivers was the key to quality and feedback. Some of them monitored the satisfaction of people with dementia and made weekly notes of activities to inform the primary caregivers. 

Safety was an issue for the initiator of the social care organization. However, unlike most initiators of the care organizations, she allowed the people with dementia to visit the city farm accompanied by a volunteer instead of a care professional.

The initiator of the neighborhood initiative indicated that they are an organization of volunteers. For him, it was important that the volunteers enjoy their work and not to bother them with protocols or official job descriptions. The initiator was establishing a collaboration with a social care organization to get support when problems would arise in the provision of day activities for people with dementia. 

The differences in ideas about quality and the need for protocols mirrored the differences in ideas about the competences and background of employees. All initiators indicated that employees should, generally speaking, be empathic, communicative, patient, and be good listeners and organizers. In line with the ideas about quality and safety issues, the respondents from nursing homes and care organizations placed emphasis on the professional background of the staff, like nursing or day activity.
“People who apply here have to have a background in care or they will be schooled in it.” 

The other respondents placed a greater emphasis on their employees’ life experience, and their affinity for working with elderly people with dementia and for working in a green environment. Some of the social entrepreneurs preferred employees without a nursing background, to prevent the nature-based ADSs from becoming what they call a “classic caring environment”. Some stressed the importance of hiring people with creative minds who are open to making a genuine connection to nature
“I work with artists, not necessarily because they are artists, but because they can interact with people from their original minds”

### 3.3. Challenges Related to Initiating and Implementing Nature-Based ADS

Based on the interviews, both general challenges, as well as more specific challenges, for the different types of initiatives were identified.

#### 3.3.1. General Challenges

All initiators, irrespective of their background, stressed that entrepreneurial behavior is a key factor in realizing an ADS in an urban area. It involved networking, being creative, having guts, taking risks, strong motivation, and perseverance. In addition, they indicated that it is important to have knowledge about financing, dementia care, and running a project.
“It demands a lot of creativity, the development and the use of a network is very important, making and keeping yourself visible, social skills and accept that it takes more time than you like”.

A second challenge is the scarcity of green spaces, especially in the city centers, which means that working together with existing green facilities can be an attractive option. Some of the initiators mentioned that it was a challenge to persuade people with dementia to participate in activities and go outside when the weather is bad. The winter is the most challenging period for outside activities. Many initiators indicated that the funding for day services was decreasing while at the same time support needed by people with dementia was increasing. Some of the work was not funded and many activities were only possible with the help of volunteers.

#### 3.3.2. Specific Challenges

Initiators encountered different types of challenges in setting up and implementing nature-based services (Table 1).

##### Social Entrepreneurs Offering Nature-Based ADS

The social entrepreneurs indicated that a key issue is to find a suitable location. Establishing a new location and developing the nature-based day service independently requires considerably investments in time, financial resources, e.g., for building, equipment, and perseverance. Using an existing green location requires clear communication with the location’s management and with existing users, like people from the neighborhood and schools. Possible drawbacks of using an existing location are that—due to the fact that the location is also used for other purposes—space may be limited, while management, or the municipality, may have other plans for the location, which may limit the use of the location by people with dementia. 

Several social entrepreneurs recognized that the nature-based ADS were very much dependent on them. They were looking for ways to make employees more responsible and stimulate their entrepreneurship. One of the initiators recognized that the project went from a pioneering phase to a stabilization phase, at which point he stepped back and appointed a manager and a person responsible for the dementia program. 

##### Nursing Homes Opening Their Gardens

The main challenges for the initiatives was to involve people with dementia who are living at home and to establish interaction with the neighborhood. Other challenges that were encountered include the commitment among employees and support within the organization, continuation of the project when the funding ends, or the project leader moves to another position, the costs for maintenance of the garden and making sure there is sufficient expertise to manage the garden and involve people with dementia in nature-based activities.
“As I said at the beginning, the enthusiasm early on is fantastic, but it is harder to keep that up. That really is a problem and not only financially, but also due to the turnover of staff. The lack of knowledge, not knowing exactly what you can do with it, connection of an experience that is not always great.” 
“In that respect, I am really glad that IVN [nature education organization] is backing me up, because they can provide us with a lot of knowledge. Real knowledge of what is already being done in this area …. So you don’t have to reinvent the wheel.” 

##### Social Care Organization Initiative

The social care organization mainly found it hard to find volunteers who are interested and able to take responsibility for people dementia visiting the city farm. A restriction was that the nature-based day service should be close to the day center of the welfare organization to allow people to get there, and the volunteer or the organization should arrange transportation to the nature-based day service.
“And finding good volunteers isn’t easy. And affinity with the target group as well as the outdoors aspect, that combination, those are unique people.”

##### Community Garden Set Up by Citizens

For the citizens initiative, getting support from the municipality to realise a community garden, was a challenge. Other organizations were also interested in using the location for other purposes. One of the key initiators had a medical background and good connections with local policy-makers and the municipality, which was helpful in securing support from the municipality. A major challenge for this initiative, run by volunteers, was to get competent volunteers involved who were willing to invest time in the initiative and who were also able to build a good relationship with people with dementia participating in the garden. Another issue was to secure the back-up and support from an organization with expertise in dementia care.

##### Hybrid Initiatives

Hybrid initiatives required committed managers of different organizations. Challenges they had to deal with included combining different sources of funding and taking differences in interests and work culture between organizations into account. According to the initiators, collaboration between organizations leads to a larger focus on the outside world, making the nature-based ADSs more attractive to a diversity of people willing to participate. Also, in these cases, the care organizations involved green experts to match green activities better with the demands of different types of participants
“Look for cooperation with many partners, schools, volunteer organizations, etc.; go outside, don’t be part of a nursing home. That makes you much more attractive to volunteers; create a lively atmosphere, etc.” 

All initiators of care organizations emphasized the importance of developing support and ownership among the employees. Major risks were a loss of enthusiasm and commitment when processes were complicated and that realization of the ADS took more time than expected or employees that decided to get a job elsewhere. 

In addition, a major challenge mentioned by the respondents was the high workload of care employees. It was often difficult to persuade them to participate, because they saw the nature-based activities coming on top of their normal tasks.
“Getting the staff on board is an issue: caregivers feel that their day is full as it is, and now they want us to do this as well.”

A specific example of collaboration was the joint initiative involving a social entrepreneur and a care organization. Initially, working together the start-up phase was beneficial. The development plan only allowed for an initiative with agricultural activities. The manager of the care organization had visited some green care farms and wanted to initiate a farm with diverse agricultural activities near the organization. Because of their shared ambitions, the farmer was hired by the care organization to provide nature-based day activities. For a few years, this was a satisfying collaboration for both partners. This changed however. When the financing of the day activities decreased, the management decided that it was too costly to hire a farmer and decided to make their own personnel (carers and day activity supervisors) responsible for the nature-based day service.
“After 2 years, cutbacks were made in healthcare. After that, they had less budget to hire me. Then there were problems in providing good care, because there were changes in personnel. The natural benefits of the collaboration, as well as just taking people to the land gradually faded out.”

According to the manager of the care organization, it was also due to the fact that the farmer was focusing too much on his agricultural ambitions, and not paying enough attention to care for the people with dementia. In her view, the project had to leave the pioneering phase and needed another impulse.

Based on the interviews with the two initiators (the care organization manager and the social entrepreneur), it became clear that these initiators had different views about the future of the nature-based day service. 

The social entrepreneur was unhappy with the attitude of the employees of the care organization who took over the supervision of the people with dementia taking part in the nature-based day service. He found it important to focus on the patients’ participation in the work that had to be done and on their interaction with people from the neighborhood growing vegetables. In his view, the care organization’s employees had little affinity with nature-based activities and tended to keep the people with dementia inside. He also mentioned that these changes started with a change in management at the care organization, after which the new manager had no affinity with the nature-based day service.
“With volunteers, you get a lively conversation and exchange. What happens now is that old people sit with old people. That wasn’t the case back then. They mixed up, old people sat with younger people and kids were walking around. I think it was much more lively for them.”
“The new manager just found it very messy, and felt that the room was dirty, so now, what they focus on is all kinds of safety regulations and whether the hygiene is in order.”

The manager of the care organization expressed her satisfaction that the nature-based ADSs was still running and that many people with dementia took part in and appreciated the activities. She confirmed that, with the changes in management, support for the nature-based ADS was no longer self-evident and that some managers were critical. 

This shows that one of the challenges for this mixed initiative is dealing with the differences in views and expectations of the care organization management and the social entrepreneur, and the different attitudes among the employees of the care organization and the social entrepreneur regarding the supervision of people with dementia. 

## 4. Discussion

Although different types of nature-based locations have been identified in urban areas, which can be beneficial to specific client groups [8,21], so far, the use of nature-based locations in urban areas to provide services to people with dementia has received little attention [10]. Nature-based services in urban areas that have been described are urban green spaces in care facilities [10].

The first aim of our study was to get an overview of the number, diversity and characteristics of nature-based ADSs in urban areas for people with dementia living at home, which led to a preliminary typology. In line with a previous study of nature-based services in rural areas [13], it is possible to distinguish different types of nature-based ADSs in urban areas for people with dementia based on the initiators’ background. This study showed that the different types of nature-based services that have been established in urban areas used different types of existing green locations such as city farms and community gardens, where other activities took place and nature-based services for people with dementia was an additional activity. Other services took place at green locations that were developed specifically to provide day activities. The characteristics and the activities of the nature-based day services and types of participants using the nature-based day service varied between the different locations and depended on their specific conditions. Nursing home gardens were mainly used by people with dementia living in the nursing homes and only by a limited number of people with dementia living at home. The focus was people being outside and enjoying the green environment. City farms and community gardens were used by a more general public, which means that people with dementia could interact with visitors and people from the neighborhood. 

The second aim of this study was to gain insight into the challenges associated with developing and running different types of nature-based ADSs for people with dementia in urban areas. 

The findings are in line with previous studies indicating that the motivation, commitment, and entrepreneurial behavior of people is crucial to the initiation and implementation of innovative services in dementia care [16,21]. Although the various types of nature-based services were appreciated by people with dementia and their primary caregivers [11], it is often a challenge for initiators to develop nature-based services in urban areas, due to a lack of green space and the difficulties involved in connecting different domains and interests [22].

The initiators used different strategies to establish nature-based ADSs. Some chose to remain independent and started their own location, while others decided to work together with existing nature-based ADSs, or to pool their resources with other initiators in a joint/mixed initiative. These choices reflect two major strategies in dealing with uncertainty in the environment: bridging and buffering. In the case of bridging, an organization establishes relationships with external stakeholders on which it depends, while, in the case of buffering, it tries to keep external stakeholders at a distance [23,24].

This study showed that the multifunctional use of existing locations, such as city farms and community gardens, including nature-based day services for people with dementia, is possible and promising, as it offered a community-oriented setting where people with dementia can interact with other citizens. One of the challenges was related to the different demands from different users and the many visitors of green locations, as a result of which the nature-base day service for people with dementia could only be offered a few days a week on such locations. 

As far as the initiatives by care organizations are concerned, it was found that, in line with previous research [17,18], leadership and managerial support were important to the implementation of nature-based day services in a care organization, while time constraints and workload were important barriers. In addition, it was found that access to green expertise and competent volunteers were important to establishing nature-based activities.

Previous studies showed that collaboration between different partners can be challenging [25]. The respondents in our study indicated that collaboration between care organizations and other institutional partners was successful and they did not indicate major challenges, unlike the collaboration between the care organization and social entrepreneur, which, though initially successful, stopped after the responsible manager was moved to a different position and financial problems motivated the care organization to spend less on the nature-based day ADS, after which there was less management commitment and differences in views became apparent. 

In line with previous findings [18], our study showed that care organizations tried to find a balance between two rationales, one of which focused on risk reduction and was in favor of hiring people with a professional care background, while the other focused on giving freedom to people with dementia and creating a setting that was as normal as possible. Nursing homes opening their garden to people with dementia living at home tended to primarily focus on risk prevention. Also, because their employees were trained and experienced in caring for vulnerable people with dementia, they focused on preventing risks. Such a care-oriented setting makes it challenging to initiate diverse and stimulating activities in which people with dementia can actively take part and it makes it difficult to involve people with dementia living at home, volunteers, and people from the neighborhood. The interviews showed that the continuity of the nature-based ADSs for people with dementia living at home was not guaranteed, but that it depended on project subsidies and the commitment of the initiators. The nature-based ADSs initiated by care organizations working together with other organizations provided more diverse activities in which people with dementia could take part actively. These more community-oriented nature-based services seemed to integrate several important aspects of dementia care like interaction with natural elements and animals, being outdoor, social interactions, meaningful activities, and person-centered approach of staff that were also found on green care farms [26]. These community-oriented initiatives faced fewer problems in terms of involving people with dementia living at home and volunteers, most likely thanks to the non-institutional society-oriented setting being created, which is confirmed by the experiences of the social entrepreneurs. In such a setting, people with dementia can interact with different types of participants and visitors. These initiatives showed the importance of involving people with expertise in green activities for people with dementia and maintenance of the green area. 

Collaboration between different types of organizations and involvement of social entrepreneurs in creating nature-based services in cities can be beneficial to innovation in dementia care [26]. The diversity of initiatives with differences in setting, expertise, focus, and type of participants, make working together a promising prospect in order to create a chain for people with dementia. The diverse and more physically challenging green activities of nature-based day services initiated by social entrepreneurs or neighborhood citizens can be a good match for people in the first stages of dementia, while the more institutionalized alternatives can be a better match for people in later stages of dementia. In addition, the initiatives of care organizations may facilitate the continuation of nature-based day activities when people with dementia move to an intramural setting. Access to nature-based services in urban areas and participation of people with dementia in nature-based activities can be important elements of healthy city projects and healthy ageing approaches.

## 5. Conclusions

Five different types of nature-based ADSs in urban areas for people with dementia living at home were identified in this study. Nature-based ADSs were developed by social entrepreneurs, nursing homes, social care organizations, and citizens. They faced general challenges such as lack of green space and funding. Other challenges were more type specific, such as the commitment among employees and limited viability of volunteers. Where most initiators established community-oriented nature-based services, nursing homes established care oriented services. Nursing homes focused on risk prevention and their nature-based services tended to be less diverse and stimulating for people with dementia living at home than the other initiatives. Collaboration between nursing homes with other types of organizations or social entrepreneurs can lead to more appealing community-oriented nature-based services.

## Figures and Tables

**Table 1 ijerph-16-01337-t001:** Characteristics of the 15 nature-based day services for people with dementia (PwD) in urban areas

Type	Initiator	Type of Location	Number of PwD/Week and Diversity Participants	Interaction PwD with Other Participants *	Main Activities	Challenges
1a	Social entrepreneur, own location	Nursery City farm City farm Greenhouse Farm	8/day 5 days/week 8/day 5 days/week 3/day 5 days/week 8/day 5 days/week 30/day 5 days/week	++ ++ ++ + +	Growing vegetables, taking care of animals, walking, preparing meals	Investments
1b	Social entrepreneur making use of existing location	Community garden City farm	10/day 1 day/week 6/day 2 days/week	++ ++	Growing vegetables, Preparing meals	Different wishes, neighborhood/visitors, and people with dementia Facilities (building, refuge) Continuation of location
2	Nursing home	Garden of a nursing home	12/day 5 days/week <1/day < 1 day/week 0	0 0 0	Enjoying the garden, Presentations, visits, excursions	Support employees Lack of volunteers Nature-based knowledge Participation of people with dementia living at home
3	Social care organization	City farm	3/day 1 day/week	+	Feeding chicken and rabbits, drinking coffee	Volunteers Nature-based knowledge Activation of people with dementia
4	Citizens	Community garden	3/day 1 day/week	++	Experiencing the garden	Expertise dementia
5a	Hybrid initiative: Care organization and other institutional partners	Community-based service: garden, greenhouse	10/day 1 day/week 12/day 5 days/week	++ ++	Gardening, growing vegetables, wood crafting	Support organization Nature-based knowledge
5b	Hybrid initiative: Care organization and social entrepreneur	City farm	12/day 5 days/week	++ - +	Shift from agricultural activities to experiencing	Support organization Cultural differences Different views and ambitions

* 0 = No interaction; + = Passive interaction (watching); ++ = Active interaction (meeting, working together).

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
