# Peer review of "Characteristics and Challenges for the Development of Nature-Based Adult Day Services in Urban Areas for People with Dementia and Their Family Caregivers"

_ijerph, 2019, doi:10.3390/ijerph16081337_

Round 1
Reviewer 1 Report
The article is interesting and gives an extended view of challenges that can be faced by careproviders to support green-based care to support people with dementia.
There are just a few speicifications that could enhance the depth of the article.
Authors should spell check and maybe consider revising some sentences, for example fist sentence in 2.3.1 "The data analysis... Corbin".
In the Method : could you speicify the profile of the different researchers that have contributed to the interview
P.4 l.163 could you specify the difference between agricultural and anture-based background ?
Results don't read so easily. I would suggest to presents the results in three big parts to make it more clear : 1. Characteristics of nature based ADS; 2. Implementation challenges, 3. General challenges.
Finally, in the discussion, to respond to findings about safety concerns and collaboration between care organisations I would suggest to add a part that takes into consideration stigma on what people are capable of doing or not, person centered care and empowerment of people with dementia. I would also discuss dementia firendly society and how green care can contribute to such a perspective, regarding the results of the interviews.
Author Response
Comments to the reviewer:
- Thank you for your positive review.
- Authors should check and consider revision some sentences e.g. the first sentence in 2.3.1. We have checked the whole paper and made quite some changes to improve the grammar of the paper.
-Methods; could you specify the profile of the researchers that have contributed to the interviews? We have added which authors contributed to the interviews.
- Could you specify the differences between agricultural and nature background. We have changed this sentence. In the previous sentence we refer to initiators with an agricultural background. The nature-based background means that they have a background in landscape architecture or nature education. We have added this information.
-Results don't read so easily. Suggestion is to present the results in three big parts. We thank the reviewer for this suggestion and have adapted the paper in this way.
-The reviewer proposes to add a part about stigma and dementia friendly society in the discussion. We have given more emphasis to the advantage of collaboration of nursing homes with community oriented organisations and social entrepreneurs as this may result in more community oriented and attractive facilities for people with dementia. We have added that the more community oriented nature-based services seem to integrate several important aspects of dementia care (e.g. natural elements, freedom of choice, autonomy, social interactions, outdoor spaces, animals, meaningful activities, person-centered approach of staff) We have not added a discussion about stigma and dementia friendly society. We think this is a step too far.
Reviewer 2 Report
General comments:
Authors should be more critical in the presentation of the results. The originality of the paper is limited by the descriptive way in which information is assembled and presented.
The paper requires some formatting revision, like missing or improper punctuation or spacing.
Specific comments:
In the methodology section the authors state that “The methodology is characterized by a qualitative, multiple case-study approach”. However, the research consists in the results of 17 interviews, therefore it is unclear how the multiple case-study approach is involved.
It would be interesting for the reader to see more information on the actual analysis.
The results section presents a long list of subsection on the characterization of the different types of nature-based ADS in urban areas and the challenges to the development of nature-based urban ADSs. Overall this section is quite unclear. Dividing in different sections the characterization and the challenges would make this section easier to read.
Most of the results focus on the characterization whilst little attention in paid to the challenges, which are the most novel part of the research. Can these be investigated more in-depth?
Is Section 3.7 a challenge? Why has it not been included in section 3.9?
3.9 should be 3.8.1 and each on the general challenges 3.8.1.1, 3.8.1.2… The same for 3.10
The discussion reads like a repetition of the results. Authors should be more critical
Conclusion should be stronger. What have you learnt from this piece of research? Is there any lesson worth passing? A framework to be developed?
Author Response
Reply to reviewer 2.
- The paper requires some formatting revision. We agree and have made changes thoughout the paper
- Authors should be more critical in the presentation of the results. We haved changed the structure of the results and have skipped some parts that were too descriptive.
- Methodology. It was stated that it is a multiple-case study approach. It is unclear how the Multi-case study approach is involved. We agree and have deleted this part and changed the general description of the study design.
- It would be interesting to see more information about the actual analysis. We have changed this section to some extent to make it more clear how the analysis was done.
- The results section is not clear. Suggestion is to divide it into sections dealing with characteristics and challenges. We agree and have adapted the structure as proposed by the reviewer.
- Can more attention be given to the challenges, which is the most novel part of the research. We think that both the characterization of the nature-based services and the challenges are interesting. We have paid quite a lot of attention to the challenges of different types of initiatives.
- Is section 3.7 a challenge? In our view quality and safety issues are not primarily a challenge. Interesting is the different views about quality and safety. We think this is such an important topic that we put it under a separate heading.
-Discussion reads like a repetition of the results. We agree that there was too much repetition. We have shortened the discussion and added some thoughts about the differences between community oriented and care oriented initiatives.
- Conclusion should be stronger. What have you learned form this research. We agree and have added our finding that collaboration of care oriented organisations (nursing homes) with other - more community oriented organisations and social entrepreneurs can lead to more appealing nature-based facilities.
Reviewer 3 Report
This is an important topic and a well written paper.
Author Response
We thank reviewer 3 for his or her positive review. We have checked the English language and grammar and made quite some changes.